# Regulation of snow-fed rivers affects flow regimes more than climate change

B. Arheimer [1], C. Donnelly [1] & G. Lindström[1]

River flow is mainly controlled by climate, physiography and regulations, but their relative importance over large landmasses is poorly understood. Here we show from computational modelling that hydropower regulation is a key driver of flow regime change in snow-dominated regions and is more important than future climate changes. This implies that climate adaptation needs to include regulation schemes. The natural river regime in snowy regions has low flow when snow is stored and a pronounced peak flow when snow is melting. Global warming and hydropower regulation change this temporal pattern similarly, causing less difference in river flow between seasons. We conclude that in snow-fed rivers globally, the future climate change impact on flow regime is minor compared to regulation downstream of large reservoirs, and of similar magnitude over large landmasses. Our study not only highlights the impact of hydropower production but also that river regulation could be turned into a measure for climate adaptation to maintain biodiversity on floodplains under climate change.

[1] Swedish Meteorological and Hydrological Institute (SMHI), 60176 Norrköping, Sweden. Correspondence and requests for materials should be addressed to B.A. (email: berit.arheimer@smhi.se)

Today's global society is dependent on water resources for sustainable development[1], but water security is under severe threat from combined pressures; human actions have become the main driver of global environmental change[2, 3]. This calls for better understanding of the cause and effect relationships and co-evolution between water resources and humans[4–6]. We may soon be approaching the planet's boundaries for global freshwater use[7] and there is empirical evidence for ongoing intensification of the water cycle due to climate change[8, 9]. For parts of the globe, however, direct human impacts on the water cycle still exceed impacts from global warming[10, 11].

A large part of the Earth's land surface receives precipitation in the form of snow. During the cold part of the year in high latitudes or high altitudes, the water is stored as snow and ice, which fully or partly melts during the spring. The seasonality in flow from such snow-fed rivers is therefore characterised by low flow during the winter followed by a high spring peak flood event. The hydromorphology, ecosystems and societies along flood-plains, lakes and shorelines in these regions have evolved over time to benefit from these flow dynamics. Examples are migratory fish, ecosystems and cultivation practices, which have evolved to benefit from the natural spring flood.

Several studies of climate-change impacts on rivers show that the annual peak flood event may be less distinct and even disappear in some snow-dominated areas[12, 13] as global warming will decrease snow fall[14] and/or the snow storage period by the end of this century[15]. More precipitation falling as rain in snow-dominated regions and shorter freezing periods will thus give less differences in river flow between seasons. Hydropower production can have the same effect on the flow regime. During spring, the river water is stored in dams and reservoirs often to be released throughout the year whenever electricity is needed most. Thus, the high flow of the snowmelt season is dampened and redistributed to other times of the year. It is known that the main drivers of change in river-flood regime include river channel engineering, land use and climate change[16] but there are knowledge gaps about their relative importance[17] and for upscaling to large domains[18]. Therefore, sufficient information on disturbance of flow regime is often missing in present assessments on ecological status for adaptation measures[19, 20].

More than 20 years ago it was noted that 77% of the river discharge from the northern part of the world is affected by fragmentation of the river channels by dams and water regulation[21]. It is recognised that this water regulation has severe effects on ecosystems and societies close to the reservoirs, for instance due to dry river channels, flow obstacles, changed flow patterns and short-term fluctuations of water level[22–24]. However, the accumulated effect on large-scale flow regime further downstream remains unknown as it is difficult to measure and separate from natural variability. Previous studies comparing climate change and regulations have therefore been limited to single reservoirs or rivers[25, 26]. In this study, on the contrary, we calculated the effects on river regimes from hydropower regulation and climate change over a large landmass.

Here, by using a detailed numerical modelling approach, we systematically quantify and compare hydropower impact with the effects of climate change across multiple rivers, from sources to the sea. We conclude that at the large scale and for floodplains in snow-dominated regions globally, hydropower regulations and climate change have about the same effect on flow regimes. Downstream of large reservoirs, however, hydropower regulations affect flow regimes much more than climate change. Overall, flow regulation should thus be key in adaptation measures for a sustainable future of snow-fed rivers and deserves much more attention by policy makers and climate-impact scientists. Our findings show that climate-change impact on flow regime is relevant in floodplains, which experience less impact from hydropower regulation (being further downstream from reservoirs). In line with the findings, we might need to reconsider the relative importance of on-going global changes and adjust adaptation measures and research accordingly.

## Results

**Hydropower regulation vs. climate change impact.** In a detailed reconstruction of natural flow regimes across Sweden, we found that current hydropower production has a significant impact on the seasonal distribution of flow, not only locally but also at the national scale (Fig. 1a). The flow peak (mean annual maximum flow) was found to be reduced by 15% and the seasonal redistribution of total river flow to the sea amounts to 19% for an average year. This number includes runoff from the whole country (also unregulated rivers) and is caused by the storage of snowmelt in reservoirs, especially in the mountains. The flow duration curve also shifts towards smaller differences between high and low flow for regulated conditions[27]. The rivers of Sweden have been exploited for large-scale hydropower productions since the early 20th century. The development of hydropower production capacity was a major contribution to the industrialisation of Sweden and amounts today to half of the electricity supply for the country, as well as an additional value in terms of meeting energy-demand peaks. There are ~1800 hydropower plants in Sweden, of which some 200 produce >10 MW, providing 94% of the total hydropower production. The total annual production varies from 50 to 75 TWh due to variability in the annual water inflows, with an average of 65 TWh/year.

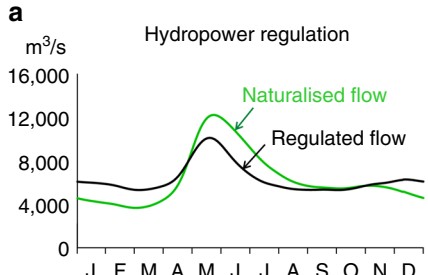

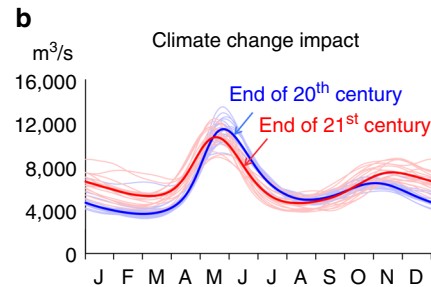

**Fig. 1** Impact from hydropower and climate change on river-flow regime for entire Sweden. Seasonal distribution of total river runoff from Sweden (450,000 km²), with and without impact of: **a** extensive hydropower regulation, and **b** projected climate change (using a climate-model ensemble of 18 members, where the mean is *bold*). Smoothed 30-yr means of daily values are shown. The climate impact modelling was based on CMIP5 projections for the representative concentration pathways (RCP) 4.5 and 8.5 from: CanESM2, CNRM-CM5, GFDL-ESM2M, EC-EARTH, IPSL-CM5A-MR, MIROC5, MPI-EMS-LR, NorESM1-M, HadGEM2-ES. Each ensemble member was downscaled using RCA[62] and bias adjusted using DBS[63]

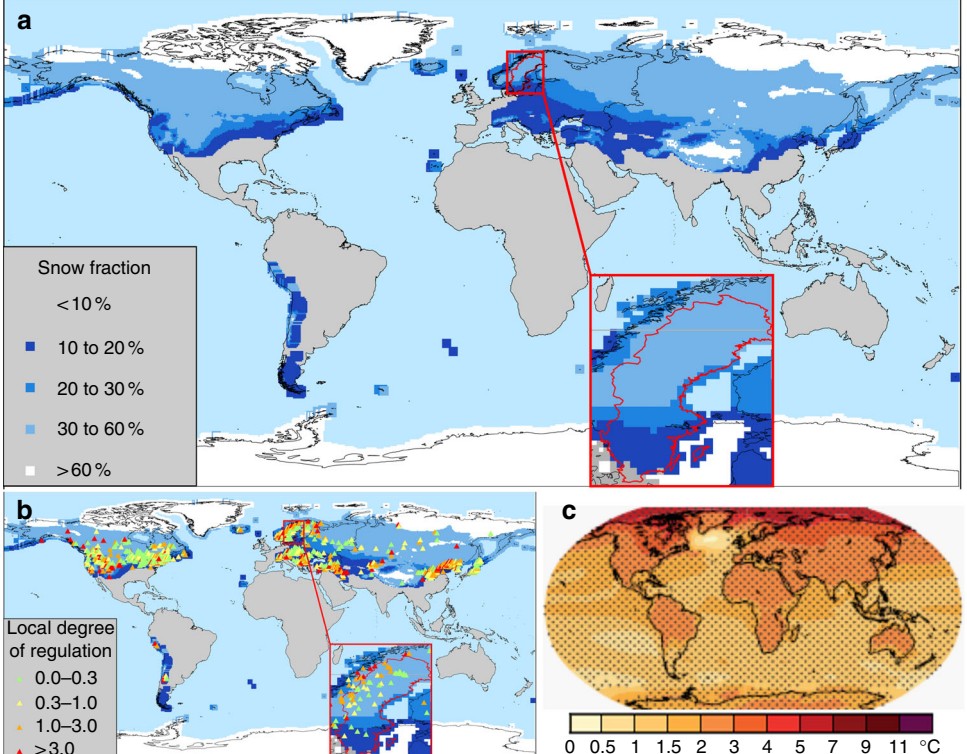

**Fig. 2** Regions with similarities in snow, hydropower regulation and temperature change. Global regions with: **a** snow fraction in precipitation, **b** the degree of regulation in hydropower dams for these regions, and **c** global warming at RCP4.5 according to IPCC, Fig. 12.11[31]. Details are highlighted for Sweden

A surprisingly similar pattern of change is found when projecting climate change impact for the same geographical domain (Fig. 1b), using a climate model ensemble with 18 members (including RCPs of 4.5 and 8.5). By the end of the century (2069–2098), 19% of the total river flow is again seasonally redistributed by a changed climate, due to less snow storage and more precipitation falling as rain. However, the flow peak is reduced by only 5% due to global warming, which gives a combined total effect for surface runoff (both regulated and non-regulated rivers) of flow peak reduction of at most 20%, if the hydropower dams are operated similarly in the future as they are today.

Additional changes to the flow regime resulting from climate change are that the spring peak starts about 1 month earlier and there will be about 10% more discharge on an annual basis. These trends have been documented in previous studies of Sweden[28–30] and this shift in timing of the snow peak due to temperature rise seems coherent across the globe[12, 13, 31]. However, the increase in annual flow cannot be extrapolated to all snow-dominated regions. These changes are caused by more precipitation in total over Sweden, but also vary spatially within Sweden with some areas getting drier[30, 32]. For parts of the snow-dominated regions, changes to the water balance may thus result in less river flows, which some authors attribute to increase in evapotranspiration[33, 34]. It should be noted that the projected climate changes in temperature, governing evapotranspiration and timing of snowmelt, are much more robust than the predictions of future precipitation (including snowfall[35]), which are uncertain and show large variability in space and time.

**Global regions of snow and hydropower.** The changes observed in Sweden are also significant on a global scale, for landmasses where river-flow generation is controlled by snowmelt,

hydropower and temperature change (Fig. 2). We assume similar water management for the snow-dominated parts of the world because hydropower production is favourable and there are similarities in climate, hydrology and energy demand. Snowmelt is stored in hydropower reservoirs, to be released at other times of the year. The snow-dominated part include mountains, which have the best energy potential for hydropower (most precipitation and head) where the snow storage thus contains a lot of accumulated energy. We found the fraction of precipitation falling as snow to be indicative of flow regime changes due to hydropower (see Methods section). The country of Sweden (450,000 km²) in Northern Europe, represents regions with 10–60% of the precipitation falling as snow. Globally, we found that a large part of the Earth's land surface also has 10–60% of precipitation falls as snow (Fig. 2a). Hence, the river regime in these areas is controlled by snowmelt as in Sweden. In these snow-dominated regions of the world, the river flow is regulated by some 2200 major hydropower reservoirs[36] (Fig. 2b) according to global data, while the actual regulation (including small dams) may be much higher. Sweden has an average or below average degree of regulation (i.e., altered capacity to store the water runoff, see Methods section). Most of Northern USA, Canada, Europe and some isolated areas of the Asian continent have similar degrees of regulation as Sweden.

Regarding climate change, the largest temperature rises are expected in the Northern hemisphere (Fig. 2c). The temperature in snow-dominated regions is projected to rise by 2–4 °C by the end of the century, assuming stabilising green-house gas emissions (RCP = 4.5)[37]. The projected temperature increases in Sweden are similar to the projected average temperature rises for other snow-dominated regions according to IPCC. In accordance with previous findings, we assume that increasing temperatures are more important than precipitation changes for snowpack seasonality in a changing climate[38]. In summary, the conclusions

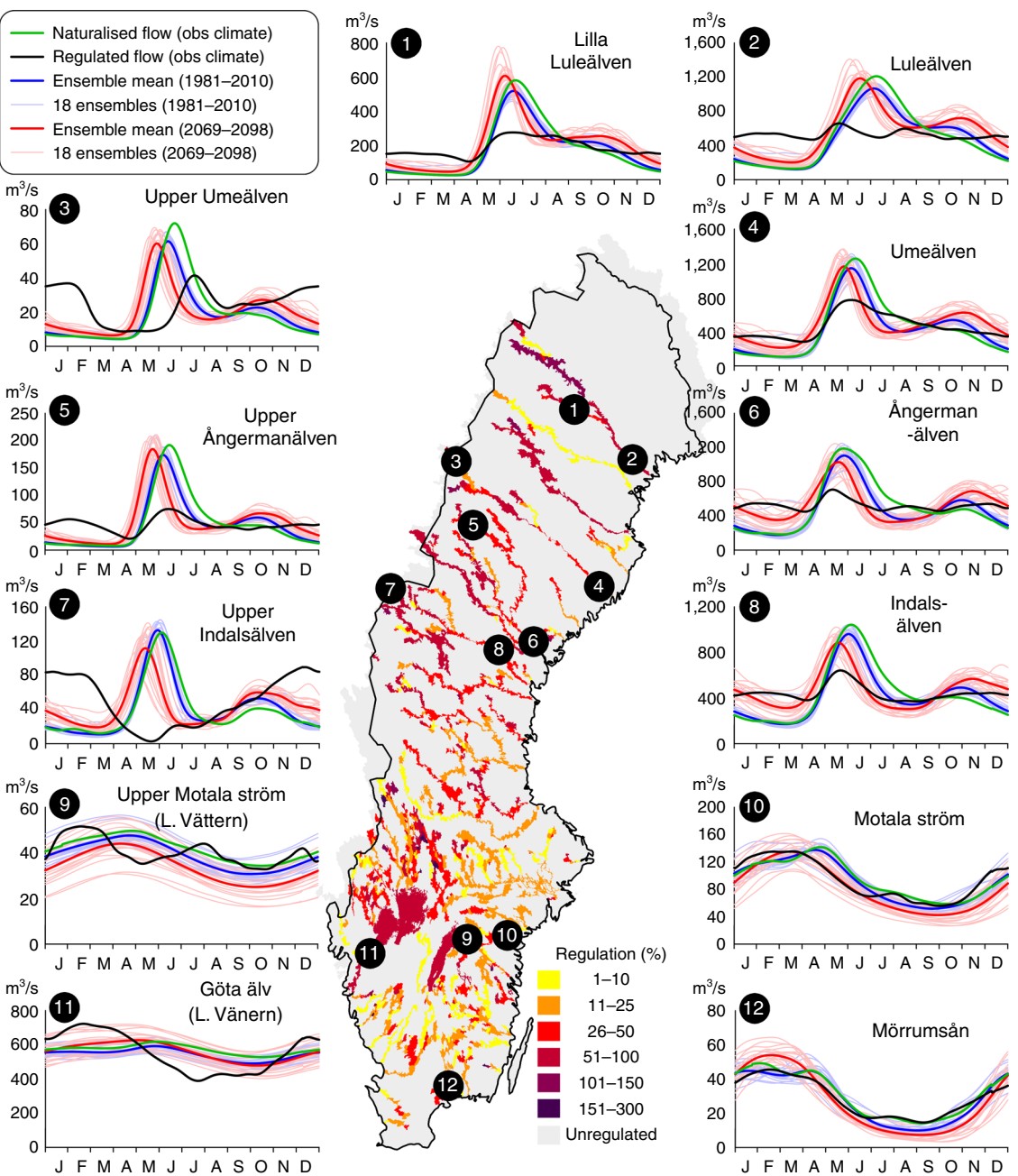

**Fig. 3** Spatial variability of impact from regulation and climate change on local flow regime. Seasonal flow regimes in 12 regulated Swedish Rivers under present conditions (*black line*) and under naturalised conditions with observed climate (*green*) or projections from climate models (*blue* and *red*). Smoothed daily mean values are shown for 30-yr period and the sites have an average regulation degree of 48% (range 25–75%). The map shows the calculated accumulated degree of hydropower regulation in Swedish rivers. The climate impact modelling was based on CMIP5 projections for the RCP 4.5 and 8.5 from: CanESM2, CNRM-CM5, GFDL-ESM2M, EC-EARTH, IPSL-CM5A-MR, MIROC5, MPI-EMS-LR, NorESM1-M, HadGEM2-ES. Each ensemble member was downscaled using RCA[62] and bias adjusted using DBS[63]

from the detailed modelling of Sweden are deemed representative for regions of snow and hydropower globally, when comparing changes of the natural flow regime caused by global warming and river regulation, respectively.

**Spatial variability and site-specific changes**. In the more detailed analysis, we found that the effects of hydropower regulation on flow regimes vary spatially and that this spatial variation can be linked to the main processes controlling river flow in different regions. At the local scale, the snowmelt peak vanishes often

completely by regulation (Fig. 3) and these rivers are associated with dry reaches and time-spells without any river flow, also at the time of the snowmelt (e.g., site No. 7 in Fig. 3). The river response to regulation shows similarities and dissimilarities across the country and can be categorised into four distinct regions, as follows.

The first region is the rivers in the mountains (northwest), which show most radically changed flow patterns with some completely reversed regimes, e.g. the Upper Indalsälven River (no. 7 in Fig. 3). These reservoirs have relatively small drainage areas and often regulation degrees of more than 100%,

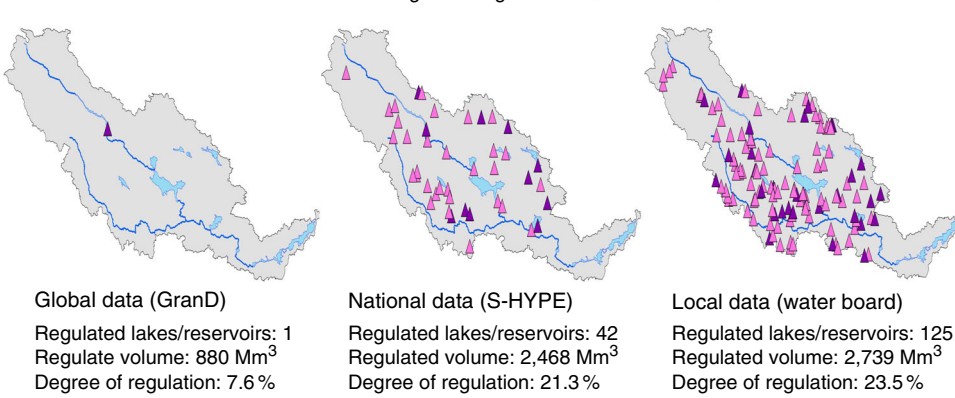

Reservoirs with local degree of regulation: ▲ 0 – 30 %  ▲ 30 – 100 %

Global data (GranD)
Regulated lakes/reservoirs: 1
Regulate volume: 880 Mm³
Degree of regulation: 7.6 %

National data (S-HYPE)
Regulated lakes/reservoirs: 42
Regulated volume: 2,468 Mm³
Degree of regulation: 21.3 %

Local data (water board)
Regulated lakes/reservoirs: 125
Regulated volume: 2,739 Mm³
Degree of regulation: 23.5 %

**Fig. 4** Information of hydropower regulation in Dalälven River using different data sources

which means that more than a year's discharge can be stored and released according to energy demand. The natural peak flow from snowmelt has disappeared completely as regulation controls basically all river flow.

The second region is the area downstream of the mountains at the floodplains near the outlet to the sea (east and southwest) where the rivers show less radical changes in flow regime, as there is a contribution also from unregulated discharge with natural peak flow. One exception is Luleälven River (No. 2 in Fig. 3), which has a total regulation degree of 67% at the outlet and an almost constant river flow over the year.

The third region includes large lakes (No. 9 and 11 in Fig. 3), encompassing the largest and 6th largest lakes in the EU. The lakes control the flow by naturally damping and storing much of the peak flow from snowmelt. A more variable flow regime has thus been introduced by regulation of these large lakes, which normally would show rather constant flow.

The fourth region is the southern plains with an annual snow fraction of only 10–20%, which show less pronounced snow peaks also under natural unregulated conditions[30]. The regulation of rivers here is mainly linked to natural lakes, which already control and dampen the seasonal flow peaks (No. 10 and 12 in Fig. 3). The seasonal change in river flow from both climate change and regulation is thus low in this region.

This example of spatial analysis in Sweden helps us to identify the regions that are more or less influenced by climate change and hydropower regulations, respectively. Such mapping can help decision makers to allocate measures for climate adaptation where they would be most effective. Climate adaptation should be targeted to where it can make a difference, and for instance floodplains are identified as areas retaining relatively high ecological status in a European perspective, worth protecting[20]. Hydropower regulation could be used in these areas for artificial flooding of floodplains to secure biodiversity in a future climate. The spatial mapping also identifies rivers where there are other potentials for improvements, for instance through more collaborative regulation strategies or rules to use the water resource for many purposes, including biodiversity aspects. Thus, different adaptation strategies in different rivers or river reaches will help us to better design the measures needed for sustainable development.

**The degree of regulation is often unknown**. We have shown that hydropower regulation radically changes the river regime for the whole surface of Sweden and conclude that similar effects are likely in other snow-dominated parts of the world, which is in line with reports from regulated and snow-fed rivers on other continents[39–41]. Only few and not very detailed assessments

on the hydrological impact of reservoirs exist on the global scale[11, 42, 43], because the degree of regulation is normally not well documented in a transparent way and local water management remains unknown, especially in open national and global databases. Hence, there is currently a knowledge gap in understanding the impact of this factor on large-scale river flow and in scientific analysis on global change. For instance, when going from the global database GranD[36] to the national database of S-HYPE[44] for Dalälven River in Sweden, the number of regulated lakes and reservoirs shifted from 1 to 42, which increased the degree of regulation by a factor of three (Fig. 4). An even more detailed local database further raised the degree of regulation from 21 to 23% when including many small constructions. The implications of this are that regulations are neglected in most large-scale assessments of climate change impacts on water resources[45–47], or that reservoir alterations are simulated with constant outflows[11, 48], which are not representative for dynamics of hydropower regulation. More attention must thus be put on documenting and sharing information on reservoir regulation and including these processes in large-scale modelling studies, to better judge their relative impacts on water security in a global change context.

## Discussion

Our findings clearly demonstrate that the common assumption of pristine hydrological conditions leads to wrong conclusions regarding on-going global changes and the impacts on large-scale river flow. We therefore show the benefits from using dynamic models that integrate both climate variability and detailed reservoir regulation. Climate change is not the main driver but regulations have significant control of river regime in snow-dominated regions, not only locally but also at the landmass scale. The ignorance among climate and hydrological scientists is because the degree of regulation is normally not well documented, kept secret or considered difficult to simulate. Neglecting or underestimating the degree of regulation will unconditionally lead to wrong conclusions when analysing global-change impact on large-scale river flow. Our results thus imply that scientists should be very careful when estimating changes to future river flow regimes until regulation can be properly addressed in the analysis. We thus urge for more complete and open global databases on flow regulation, and water management for integrated and detailed modelling at the global scale. New techniques using satellites and crowd sourcing could also be helpful here.

Among fresh-water ecologists, on the other hand, the impacts of severe changes in river regime from regulations are well documented[23, 49, 50] and widely discussed also in a climate-

**Table 1 Model skills in predicting hydropower regulation**

| River | Hydropower plant (dam) | Recharge area (km$^2$) | Upstream lakes (%) | Flow regulation (%) | NSE QR | NSE QN | NSE ΔQ |
|---|---|---|---|---|---|---|---|
| Luleälven | Seitevare | 2,250 | 7 | 85 | 0.29 | 0.64 | 0.69 |
| Luleälven | Boden | 24,924 | 9 | 67 | 0.07 | 0.88 | 0.76 |
| Umeälven | Stornorrfors | 26,568 | 8 | 25 | 0.82 | 0.93 | 0.73 |
| Ångermanälven | Sollefteå | 30,638 | 9 | 37 | 0.70 | 0.91 | 0.86 |
| Indalsälven | Hammarforsen | 23,842 | 10 | 39 | 0.63 | 0.90 | 0.84 |
| Motalaström | Motala | 6,384 | 35 | 65 | 0.31 | 0.71 | 0.14 |
| Motalaström | Holmen | 15,384 | 21 | 41 | 0.79 | 0.89 | 0.09 |
| Göta älv | Vargön | 46,886 | 19 | 74 | 0.70 | 0.91 | 0.46 |
| Average | | | | | 0.54 | 0.85 | 0.57 |
| Median | | | | | 0.66 | 0.90 | 0.71 |

S-HYPE model performance estimated by the NSE[60] criteria at eight hydropower plants, using daily values for river flow including regulation (QR) tested against observations; for naturalised conditions (QN) tested against independent reconstruction; and for the hydropower impact (ΔQ) tested against observations combined with independent reconstructions. (From Arheimer and Lindström, 2014.)

change context[19, 51, 52]. Ecosystems in regulated rivers are considered more vulnerable to climate change[43, 53] but also more favourable for adaptation measures as flow regimes can be manipulated[25, 54]. Our detailed mapping of flow regulations for a large landmass indicate that the radical change from natural flow regime in mountains would be difficult to restore. We therefore recommend more attention to downstream areas and floodplains that receive water flow also from unregulated parts of the river network. In these areas, we found that climate change will have about the same impact as hydropower regulations. Hence, climate change will have more severe consequences on present status of biodiversity in floodplains, and it might be worth to introduce artificial flooding for climate adaptation in these regions. The regulations could thus help in climate adaptation, but there may be high costs for energy loss and melt-water must still be available in sufficient amounts from snow storage.

The hydropower sector is also subject to future change. It is not yet known how climate change will impact the regulation schemes for power production, as timing of both water supply and energy demand changes. With lower spring flows, reservoirs may need less storage capacity and would thus affect natural flows less. On the other hand, electricity demand may also change over time and reservoir storage may be used to balance out fluctuations in other renewable power sources, such as wind and solar.

Despite its side effects, hydropower is referred to as a clean and renewable energy source, which is favoured over fossil fuels. The growth in new hydropower projects has currently moved to countries with emerging economies[55]. This might be challenging as water governance require collaborations among multiple partners to ensure domestic, industrial, agricultural or environmental uses[10, 56, 57]. In Sweden, collaboration has developed over the decades between various hydropower companies along the rivers, to better harmonise regulation schemes and improve interactions with government authorities. This is a good role model; however, it should also be recognised that some countries may not have the economic, legal or political capacity to implement such governance. The global community will then be crucial to support the UN Paris Agreement and the UN sustainable development goals.

## Methods

**Simulating change in river flow.** The impacts of change in flow regime caused by hydropower regulation and climate change, respectively, were estimated using the Hydrological Predictions for the Environment (HYPE)[58] numerical model. The HYPE model is a process-oriented integrated catchment model, which is continuously released in new versions for open access at http://hypecode.smhi.se/. This model has been applied at the large scale for several parts of the globe (http://hypeweb.smhi.se/) and the set-up for Sweden is called S-HYPE[44]. We used dynamic model routines to predict river regulation and naturalised flow,

respectively, and forced the model with meteorological variables from a 4-km grid, either based on optimal interpolation of observations[59] or an ensemble of downscaled climate projections (from http://www.cordex.org/). In both cases, we used daily values for the reference period 1981–2010 to evaluate effects of change. The total effect of redistribution of flow between seasons was calculated by comparing 30 years averages for each day between naturalised flow for the reference period with 30 years averages for each day during river regulation and climate change conditions, respectively.

S-HYPE[44] is a national multi-basin model system for Sweden that covers more than 450,000 km$^2$ and produces daily values of hydrological variables in 37,000 catchments from 1961 onwards. The spatial resolution is on average 10 km$^2$ and it covers the Swedish landmass, including transboundary river basins with Norway and Finland. The model is used operationally for water management and the national warning service for floods and droughts. Most catchments are ungauged, but observations are available in 400 sites for model evaluation of daily water discharge and 86% of the river flow from land to the sea is monitored. A number of model-performance criteria are estimated in each site, e.g., Nash and Sutcliffe efficiency (NSE)[60] and relative error. The latest S-HYPE version (2012) has on average daily NSE = 0.83 for 222 stations with ≤ 5% regulation and an average relative volume error of ±5% for the period 1999–2008. For all gauging sites with both regulated and unregulated rivers, the mean monthly NSE = 0.80. Average NSE includes catchments ranging from a few to several tens of thousands of km$^2$ and various land-uses across the country. The S-HYPE model provides different kinds of water information and open data to Swedish water authorities and the public, free to download from the web site: http://vattenwebb/. The model system is also used in scenario simulations to describe changed conditions.

The method to predict regulated flow (QR) made use of current approach to model regulation in S-HYPE. The model set-up includes 509 regulated lakes and reservoirs, and 23 man-made river diversions leading water over catchment borders. Each regulated reservoir or group of reservoirs is treated separately, with individual storage volumes as input data. The model simulates the alteration of river flow in a conceptual way by water storage from spring and summer to hydropower production during autumn and winter. The seasonal production pattern is estimated individually from observations of discharge and water levels. This was done explicitly for some 50 gauged dams, and group-wise for some 400 lakes and reservoirs upstream of river gauges. Some small dams are modelled by using a general regulation routine[27] with the following function: (i) when the water level is low production is reduced, (ii) at moderate water levels the outflow only depends on the time of the year, (iii) when a dam is nearly full, discharge occurs through the spillways. The spillway flow is modelled by a rating curve, which is calibrated separately using the same observations as when estimating the seasonal production.

When evaluating the method for predicting impact from hydropower production, the routine of flow regulation in S-HYPE resulted in monthly average NSE = 0.69 for the 176 gauges with >5% degree of regulation. Reservoir regulation is often very variable on a daily basis, and therefore, monthly NSE is relevant for judging model performance for flow regime.

The Method to predict non-regulated and naturalised flow (QN) made use of current approach to model lakes in S-HYPE. The model set-up has 9082 non-regulated lakes explicitly modelled at sub-basin outlets. Lake routing is modelled by establishing rating curves from observed discharge and lake-water levels. These are either explicitly determined from observations (from various time-periods) in individual lakes, calibrated group-wise using downstream gauges or for regions, or by using a general rating curve[61]. When simulating non-regulated conditions, assumptions about such natural rating curves for original lake outlets must be made for sites with lake regulation today. For 30 major reservoirs, we established a specific rating curve to describe naturalised flow based on measurements of water discharge and lake level fluctuations, either by observations

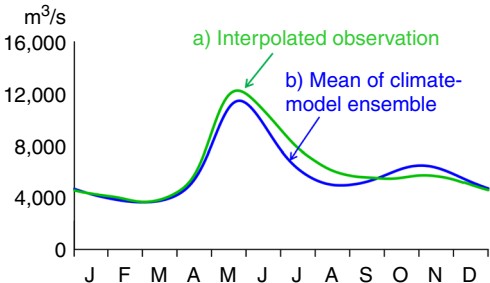

**Fig. 5** Flow regime of the reference period when using observations or climate data. Daily average values of unregulated river flow from land to sea for the period 1981–2010, from the S-HYPE model, **a** using the national 4 km meteorological grid based on observations, and **b** for the mean of the 18 members in the climate model ensemble

prior to regulations or by using reconstructions made by hydropower companies, which are currently used for legal justifications in the water court. For the 476 remaining lakes, we used the equations for the spillways from regulated conditions. Naturalised flow was then modelled by using these new rating curves and removing all regulation storages and man-made diversions in the model. Three man-made lakes were removed completely and replaced with forest on till soil.

The daily effect ($\Delta Q$) of hydropower regulation on river flow was calculated as (Eq. 1):

$$\Delta Q(t) = QR(t) - QN(t) \tag{1}$$

The HYPE modelling of naturalised flow was evaluated against more detailed independent reconstructions based on observed water levels for eight reservoirs across Sweden (Table 1). All stations showed NSE >0.7, except the highly regulated Seitevare, which has a rather small drainage basin and 85% flow regulation with intense short-term fluctuations. We explicitly tested the model predictability of hydropower impact, by studying the effect itself in the HYPE-model compared to observations vs. reconstruction, there was normally a good agreement with a median NSE = 0.71. The performance was related to degree of regulation and upstream lake area. The sites with high flow regulation showed low NSE values and poor skills were also noted at the outlet of Lake Vättern (Motala), which is a very large lake compared to the drainage basin that feeds the river. The dampening of the hydrograph, higher influence of evaporation, and long-term fluctuations in lake water make it more difficult to reach a high NSE at the outlet. In addition, the outflow of Lake Vättern is more affected by short-term regulation than by seasonal re-distribution of the flow. In addition to statistical criteria, the model performance for various sites was also evaluated by plotting flow duration curves and time-series[27].

When modelling climate change impact, we used a state-of-the art modelling chain to assess the climate change impact in hydrology. The S-HYPE model was forced with transient time-series from downscaled and bias-corrected output from an ensemble of climate models for the period 1961–2100. To estimate climate change impact, the river flow at the end of the century (2068–2098) was compared with a reference period (1981–2010) for each ensemble member. We used CMIP5 projections for the representative concentration pathways (RCP) 4.5 and 8.5, respectively, from the following nine Global Circulation Models (GCM): CanESM2, CNRM-CM5, GFDL-ESM2M, EC-EARTH, IPSL-CM5A-MR, MIROC5, MPI-EMS-LR, NorESM1-M, HadGEM2-ES. Each of the GCMs was dynamically downscaled from 1000 to 50 km by the RCA model[62] version 4, as part of the CORDEX initiative (http://www.cordex.org/) and thereafter statistically downscaled and bias-corrected to the national 4 km meteorological grid based on observations[59] using the distributed based scaling (DBS) method[63].

In total, the impact from 18 climate projections were then simulated by using the unregulated version of S-HYPE to create an ensemble of projected river flows. The total flow from land to sea was compiled as well as river flow from selected rivers with less regulation. The results were quality assured by comparing results with previous estimates of climate change impact in Sweden. The S-HYPE results for unregulated flow for the reference period (1981–2010) were not identical when comparing the model forced by observations with the mean of forcing from climate models (Fig. 5). This was because the the bias correction was done for the period 1961–1990, which allowes the climate signal to differ between models (and to observations) from 1990 and onwards.

**Identifying global regions of relevance**. When discussing the relevance of our results to global scale, we assume robustness in links between changes in climate, snow fraction and peak flow, as well as between snow fraction and reservoir management. For the first, previous studies at the global scale[38] shows that warming is more important than precipitation changes for snowpack seasonality; strong decrease in winter snow accumulation and spring snowmelt was projected

regardless of precipitation changes. The same findings have been observed for Sweden[28–30] and other regions worldwide[12, 13, 31].

The second assumption implies that the snowmelt during spring is stored in the hydropower reservoirs to be released at other times of the year. This was guided by hydrological interpretation of similarities in observed flow signatures at continental scale (using 1366 river gauges), showing that all snow-dominated regions had clear influence of hydropower regulation in most hydrographs[64]. Following, we did an empirical study using observations from major reservoirs across Europe, representing a wider range of dam types, operations and climate than in Sweden and giving an indication of a possible global relationship. We compared seasonality in observed outflow at dams outlets with the seasonality of simulated inflows, using a pan-European hydrological model (E-HYPE v2.1)[65]. For each dam, we quantified the month of peak natural inflows and the month of peak regulated outflows. The difference reflects the impact of the dam on natural flow regime.

We then compared the change in peak flow month with a number of factors including mean winter temperature, dam capacity, dam capacity compared to inflows, the dam's regulation volume compared to inflows, and the fraction of precipitation falling as snow (snow fraction). The only significant relationship was found between change in peak flow month and snow fraction and ($R2 = 0.19$, $p = 0.001$). We found no significant relationship for any of the other variables. Snow fraction was thus found to be an indicator of hydropower regulation with seasonal redistribution of flow.

Snow-dominated regions were identified worldwide by calculating the average snow fraction for the period 1981–2010 from a global rain and snow data set. The WFDEI data set[66] based on the WATCH Forcing Data methodology applied to ERA-Interim reanalysis data was used, at a resolution of 0.5 degree. The information of spatial patterns of projected global temperature rise by the end of the century was taken from Chapter 12 of The Physical Science Basis. Contribution of Working Group I to the Fifth Assessment Report of the Intergovernmental Panel on Climate Change[37].

Global information of hydropower regulation was collected from the Global reservoir and dams' database, GranD[36]. This data set collates data on reservoirs with a capacity >0.1 km³, which includes more than 6000 reservoirs worldwide with a combined capacity of 6200 km³. To determine the large-scale degree of regulation we used the global composite runoff field in GranD (called GCRF), which combines observed discharge with climate-driven runoff estimates to get a composite runoff field consistent with observations. The local degree of regulation at each dam ($D_{reg}$) was calculated by dividing the dam capacity with the mean annual inflows to the dam $V_{runoff}$. This gives an indication of the dam's capacity to store the runoff generated over a year, i.e., if $D_{reg} > 1$, the dam can hold all runoff generated within 1 year, and if $D_{reg} = 0.5$ the dam can hold half of the runoff generated within 1 year. The local degree of regulation at each dam was used instead of the accumulated degree of regulation per river as the connectivity between dams on the same river basin was not available in open global databases.

**Data availability**. The data that support the findings of this study are available in Zenodo with the identifiers 'doi.org/10.5281/zenodo.581145'[67] for hydropower impact modelling and 'doi.org/10.5281/zenodo.581186'[68] for climate impact modelling. Original climate projections are available in ESGF at http://www.cordex.org/. River flow observations and catchment delineation for Sweden are available at http://vattenwebb.smhi.se/. The HYPE model code is open source and available for inspection and free download at http://hypecode.smhi.se/.

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

## Acknowledgements

The study was performed within the EU FP7-funded project SWITCH-ON (grant agreement 603587), which explores the untapped potential of Open Data to tackle changes in the Hydrosphere. Modelling of climate-change impact in Sweden was funded by the Knowledge Center for Climate Change Adaptation at SMHI and we would like to acknowledge contributions from Elin Sjöqvist and Jenny Axén-Mårtensson at SMHI for this part. Modelling of the hydropower influence was funded by the Swedish Agency for Marine and Water Management (HaV) and we would like to acknowledge valuable data of Dalälven River from Niclas Hjerdt, SMHI. The investigation was performed at the SMHI Hydrological Research unit, where much work benefits from joint efforts in developing models and concepts by the whole team. The scientific findings will contribute to the decadal research initiative "Panta Rhei—changes in hydrology and society" by the International Association of Hydrological Sciences (IAHS).

## Author contributions

B.A. contributed with the idea, the overall study design, result analysis, figures and writing the manuscript; C.D. contributed with identifying global regions of relevance, Fig. 2, and commenting on the manuscript; G.L. contributed with computational calculations, developing the S-HYPE model to represent regulated and unregulated conditions, compilation of model output, figures and commenting on the manuscript.

## Additional information

**Competing interests:** The authors declare no competing financial interests.

