## [Peer Review File · Nature Communications]

Reviewers' comments:

Reviewer #1 (Remarks to the Author):

The paper claim that regulation of snow-fed rivers results in larger river regime changes than climate change. I think it is good the authors address this issue and show data on this from Swedish rivers. I agree in many of the statements made, but the main issue with the paper is that these statement are well known and do not add much to the current scientific knowledge. Especially, the conclusion stating that "exploitation by humans provide a stronger signal than climate change" is evident and provide very limited news to the scientific community. As a useful input, the paper present evidence from global and Swedish perspectives, but the paper fail to provide details on how this adds to the existing literature on the topic. A quick literature search on google scholar show that there are many papers on the issue from the last few years. Several papers show the impact of regulation in different regions of the globe, also snow covered regions.

Reviewer #2 (Remarks to the Author):

1. What are the major claims of the paper?

The authors claim that regulation of snow-fed rivers (for hydropower production) affects flow regimes more than climate change. They use this to then claim that (i) regulation should not be omitted from climate change impact assessments (ii) climate change impact on river regime in snow-dominated areas will only be minor compared to the extensive exploitation of water resources by humans that is currently introduced or existing since many decades (iii) for water management in snow-dominated catchments, strategies for flow regulation are more important than climate change adaptation

2. Are they novel and will they be of interest to others in the community and the wider field?

The claims are novel and of interest both scientifically, and in a wider water management context, and further in climate change adaptation policy debates.

3. Is the work convincing, and if not, what further evidence would be required to strengthen the conclusions?

The authors present credible model-based evidence for Sweden that the magnitude of past change in flows due to water management for hydropower production is similar to the magnitude of projected future flow changes due to climate change.

The authors conclude that impacts of flow regulation provide an important context for climate change, and require more attention. I also found this credible.

However, I was not convinced of two other conclusions:

(a) it was not clear that the results of calculations from Sweden can be transferred to water management for other countries with snow-dominated catchments. Presenting global maps of snow-fraction, degree of regulation and projected temperature rise and stating that the ranges of these 3 indices for Sweden are similar to the global ranges is not sufficient. For this to be adequate, someone needs to establish that these three variables constitute suitable measures of similarity for water management.

(b) The authors conclude that changes caused by climate will be minor, by comparing them with past changes due to flow regulation. I find this comparison misleading, and open to misuse. What really counts is whether, over the coming century, the impacts of change in future flow regulation are smaller or larger than those of change in climate. I think the correct comparisons to make are between future climate change impacts and (i) FUTURE flow regulation changes (ii) adaptive capacity of society to future changes in water availability. While Sweden may be well-placed to adapt to the impacts of climate change on water resources, it does not follow that other countries are similarly placed. Other countries may have already altered their water resources as much as possible without additional change causing serious harm, or they may not have the economic, legal or political capacity to implement adaptive strategies. If it is globally true that climate change

impacts are generally smaller than water management impacts (I'm not convinced of that yet - see my point a), then I would like to see the adaptation response framed not just in terms of Sweden's response to climate change, but also addressing what could other countries do if they have less adaptive capacity than Sweden?

4. On a more subjective note, do you feel that the paper will influence thinking in the field?

Yes it has the potential to be used in important debates about water management. For this reason I would like to see the main points made more convincingly.

Detailed Comments

P2 "the redistribution of total river flow to the sea amounts to 19% for an average year" The meaning of redistribution is unclear. Can the authors be more specific? For example, is this redistribution in time, or in space?

P2 "additional value in terms of meeting effect peaks" what are effect peaks?

P3 "Global regions of snow and hydropower" I found this section interesting, but not sufficiently convincing that I would expect results from Sweden to be transferable to the rest of the world. I think the authors need to justify that the three indices they present are suitable similarity indices for water management.

P5 "We have shown that hydropower regulation radically changes the river regime" The authors show this in Sweden, but not in other regions. However, I am sure suitable literature could be cited for this point.

P6 "Climate change is not the main driver but regulations have by far much more control of river flow in snow-dominated regions" This argument disguises the true question for society. In deciding whether climate change is important in this context, I think it is more relevant to consider whether, over the coming century, the impacts of future changes in flow regulation are smaller or larger than those of change in climate. If instead, the authors want to make the point that Sweden has already demonstrated large adaptive capacity, and has further capacity for additional smaller changes, they need to express that more clearly.

P6 "we argue that the climate change impact on river regime in snow-dominated areas will only be minor compared to the extensive exploitation of water resources by humans that is currently introduced or existing since many decades" I don't see why this is a meaningful comparison. See my point immediately above.

P6 Surely the real question is whether society has sufficient adaptive capacity to deal with additional changes in flow regime, beyond those which have been caused by past flow regulation. I agree with the authors that Sweden has a world-leading methodology for addressing such questions. I think the authors' optimism that Sweden has the capacity for such adaptation is well-placed. Prospects seem less optimistic for some other countries, and this would perhaps be worthy of some discussion.

Response to all of the concerns raised by Reviewer #2 (in red)

Reviewer #2: 1. What are the major claims of the paper?

The authors claim that regulation of snow-fed rivers (for hydropower production) affects flow regimes more than climate change. They use this to then claim that (i) regulation should not be omitted from climate change impact assessments (ii) climate change impact on river regime in snow-dominated areas will only be minor compared to the extensive exploitation of water resources by humans that is currently introduced or existing since many decades (iii) for water management in snow-dominated catchments, strategies for flow regulation are more important than climate change adaptation **agree**

2. Are they novel and will they be of interest to others in the community and the wider field?

The claims are novel and of interest both scientifically, and in a wider water management context, and further in climate change adaptation policy debates. **agree**

3. Is the work convincing, and if not, what further evidence would be required to strengthen the conclusions?

The authors present credible model-based evidence for Sweden that the magnitude of past change in flows due to water management for hydropower production is similar to the magnitude of projected future flow changes due to climate change.

The authors conclude that impacts of flow regulation provide an important context for climate change, and require more attention. I also found this credible.

However, I was not convinced of two other conclusions:

(a) it was not clear that the results of calculations from Sweden can be transferred to water management for other countries with snow-dominated catchments. Presenting global maps of snow-fraction, degree of regulation and projected temperature rise and stating that the ranges of these 3 indices for Sweden are similar to the global ranges is not sufficient. For this to be adequate, someone needs to establish that these three variables constitute suitable measures of similarity for water management.

We have tried to improve this part to validate the significance of our results. When identifying global regions of relevance of our results, we assume robustness in links between (i) changes in climate, snow fraction and month of peak flow in the flow regime; and (ii) snow fraction and reservoir management. We have now explained the basis for the assumptions more clearly in the paper itself and in the Methods section. We have also added more references to support the assumptions. The following improvements have been made:

- **In the Result section under “Global regions of snow and hydropower”, we have changed the structure of the text to read more clearly. We have added a few**

sentences including a reference. We also start this section by clarifying why we assume similarity for water management in the snow-dominated regions.

- In the new manuscript, we further mention in the Discussion Section that growth in new hydropower projects has currently moved to countries with emerging economies – this implies that even more similar water management in the snow-dominated regions is to be expected in the future.
- In the Result section under “The degree of regulation is often unknown” we have added 3 references confirming that our results are in line with reports from regulated and snow-fed rivers on other continents.
- In the Methods section under “Identifying global regions of relevance” we have added 3 paragraphs, describing empirical findings of (i) changed flow regime in snow dominated regions at the continental scale (with literature reference) and (ii) correlation between snow fraction and change of snow-melt peak by reservoirs.

(b) The authors conclude that changes caused by climate will be minor, by comparing them with past changes due to flow regulation. I find this comparison misleading, and open to misuse. What really counts is whether, over the coming century, the impacts of change in future flow regulation are smaller or larger than those of change in climate. I think the correct comparisons to make are between future climate change impacts and (i) FUTURE flow regulation changes (ii) adaptive capacity of society to future changes in water availability. While Sweden may be well-placed to adapt to the impacts of climate change on water resources, it does not follow that other countries are similarly placed. Other countries may have already altered their water resources as much as possible without additional change causing serious harm, or they may not have the economic, legal or political capacity to implement adaptive strategies. If it is globally true that climate change impacts are generally smaller than water management impacts (I'm not convinced of that yet - see my point a), then I would like to see the adaptation response framed not just in terms of Sweden's response to climate change, but also addressing what could other countries do if they have less adaptive capacity than Sweden?

We agree that there was a risk of misusing our results (Thanks for pointing this out!).

We have therefore changed the wording throughout the paper to better address future implications for climate adaptation of freshwater ecology. We have also added more references to the scientific literature (25 new citations). Especially, we have rewritten the Discussion Chapter by raising three major implications of our results, addressing different target audiences:

1. Climate and hydrological scientists, when analysing global-change impact on large-scale river flow.
2. Freshwater ecologists and water managers, when establishing climate adaptation strategies for biodiversity.
3. Global development community, on potential changes in future flow regulations for existing reservoirs and through new constructions.

Regarding the countries with less adaptive capacity, we have now highlighted this problem at the end of the discussion section. This was very good input.

4. On a more subjective note, do you feel that the paper will influence thinking in the field? Yes it has the potential to be used in important debates about water management. For this reason I would like to see the main points made more convincingly. We have re-written parts of the paper to be clearer on our main points and implications for climate adaptation.

(FYI: We are currently investigating the measures we suggest in a numeric experiment for a Swedish case-study of the floodplain in the Dalälven River – testing the potential for using hydropower regulation for induced flooding of floodplains to secure biodiversity in a future climate. The regulations could thus help in climate adaptation, but there might be high costs for energy loss and water must be available in sufficient amounts. We make calculations for a more detailed cost-benefit analysis on behalf of the local/regional water managers)

Detailed Comments

P2 "the redistribution of total river flow to the sea amounts to 19% for an average year" The meaning of redistribution is unclear. Can the authors be more specific? For example, is this redistribution in time, or in space? **Should be 'seasonal redistribution' which is now clarified in the manuscript.**

P2 "additional value in terms of meeting effect peaks" what are effect peaks? **Should be "Energy demand peaks", which is now clarified in the manuscript.**

P3 "Global regions of snow and hydropower" I found this section interesting, but not sufficiently convincing that I would expect results from Sweden to be transferable to the rest of the world. I think the authors need to justify that the three indices they present are suitable similarity indices for water management. **We have now explained the basis for using these indices more clearly in the paper itself and in the Methods section (see Response above).**

P5 "We have shown that hydropower regulation radically changes the river regime" The authors show this in Sweden, but not in other regions. However, I am sure suitable literature could be cited for this point. **As suggested, we have inserted a number of references here.**

P6 "Climate change is not the main driver but regulations have by far much more control of river flow in snow-dominated regions" This argument disguises the true question for society. In deciding whether climate change is important in this context, I think it is more relevant to consider whether, over the coming century, the impacts of future changes in flow regulation are smaller or larger than those of change in climate. If instead, the authors want to make the point that Sweden has already demonstrated large adaptive capacity, and has further capacity for additional smaller changes, they need to express that more clearly.

We agree and we have now put more attention throughout the paper on climate change impact on present conditions (with regulations) and suggest using regulations for climate adaptation, for instance in floodplains where the impact of climate change is still of importance to flow regime. We also discuss potential changes in regulation due to climate change and future prospects of new regulations in emerging countries. However, we have not done research specifically on this, so it is only reflections linked to our results.

Sweden has adapted during the last century, but while the changes in flow regime don't impact largely on people in Sweden nowadays, they change the aesthetic value of the river, conditions for fishing and recreation and have a large impact on ecology.

P6 "we argue that the climate change impact on river regime in snow-dominated areas will only be minor compared to the extensive exploitation of water resources by humans that is currently introduced or existing since many decades" I don't see why this is a meaningful comparison. See my point immediately above. **This sentence is now removed.**

P6 Surely the real question is whether society has sufficient adaptive capacity to deal with additional changes in flow regime, beyond those which have been caused by past flow regulation. I agree with the authors that Sweden has a world-leading methodology for addressing such questions. I think the authors' optimism that Sweden has the capacity for such adaptation is well-placed. Prospects seem less optimistic for some other countries, and this would perhaps be worthy of some discussion. **Please, note that also in Sweden there are ongoing debates both on design of environmental flows at hydropower regulations, and climate adaptation strategies to implement to ensure biodiversity. However, we do want to bring forward the successful collaboration in river regulation companies.**

We have now inserted a last paragraph in the Discussion section raising the concern with new hydropower projects in emerging countries.

REVIEWERS' COMMENTS:

Reviewer #2 (Remarks to the Author):

The authors have satisfactorily addressed the issues which I raised in my previous review. The relative importance of climate change vs river regulation are now presented in a relatively balanced fashion, with relevant caveats regarding scale and presence/absence of regulation. The recommendations for using regulation to mitigate future adverse climate change impacts are now placed more appropriately in a wider context, recognising that not all countries have the same adaptive capacity to respond to change.

I have only a couple of minor additional points which relate to new text:

1. "the snow storage itself contains a lot of accumulated energy" I found this vague and unclear
2. "Sweden has an average or below average degree of regulation" This and the following sentence need to make clear how degree of regulation is being assessed.

Response to Referee N#2

I have made the following small changes to address the final comments on the manuscript "Regulation of snow-fed rivers affects flow regimes more than climate change" by Arheimer, B., Donnelly, C. and Lindström, G. - I hope this makes it clearer.

1. "the snow storage itself contains a lot of accumulated energy" I found this vague and unclear

This sentence is now re-phrased, and reads:

The snow-dominated part include mountains, which have the best energy potential for hydropower (most precipitation and head) where the snow storage thus contains a lot of accumulated energy.

2. "Sweden has an average or below average degree of regulation" This and the following sentence need to make clear how degree of regulation is being assessed.

This sentence is now re-phrased, and reads:

Sweden has an average or below average degree of regulation (i.e. altered capacity to store the water runoff, see Methods section).